


# The role of spatial dependence for large-scale flood risk estimation

Ayse Duha Metin[1,2], Nguyen Viet Dung[1], Kai Schröter[1], Sergiy Vorogushyn[1], Björn Guse[1],
Heidi Kreibich[1], Bruno Merz[1,2]

[1]GFZ German Research Centre for Geosciences, Section Hydrology, 14473 Potsdam, Germany
[2]Institute of Environmental Science and Geography, University of Potsdam, 14476 Potsdam, Germany

*Correspondence to*: Ayse Duha Metin (duhametin@gmail.com)

**Abstract.** Flood risk assessments are typically based on scenarios which assume homogeneous return periods of flood peaks throughout the catchment. This assumption is unrealistic for real flood events and may bias risk estimates for specific return periods. We investigate how three assumptions about the spatial dependence affect risk estimates: (i) spatially homogeneous scenarios (complete dependence), (ii) spatially heterogeneous scenarios (modelled dependence), and (iii) spatially heterogeneous, but uncorrelated scenarios (complete independence). To this end, the model chain RFM (Regional Flood Model) is applied to the Elbe catchment in Germany, accounting for the space-time dynamics of all flood generation processes, from the rainfall through catchment and river system processes to damage mechanisms. Different assumptions about the spatial dependence do not influence the expected annual damage (EAD), however, they bias the risk curve, i.e. the cumulative distribution function of damage. The widespread assumption of complete dependence strongly overestimates flood damage in the order of 100% for return periods larger than approximately 200 years. On the other hand, for small and medium floods with return periods smaller than approximately 50 years, damage is underestimated. The overestimation aggravates when risk is estimated for larger areas. This study demonstrates the importance of representing the spatial dependence of flood peaks and damage for risk assessments.

## 1. Introduction

Floods are frequently occurring as destructive events throughout the world. In the period 1995-2015, there were 3062 flood events which affected 2.3 billion people worldwide with overall damages of US$662 billion (CRED and UNISDR, 2015). It is commonly stated that flood risk has increased rapidly in the past and will continue to increase in future due to the combined effects of climate change and socio-economic development (e.g. Rojas et al., 2013). In order to mitigate the destructive impacts of floods, sound flood risk assessment and management are essential.

During the last decades, flood risk management has gained considerable attention and has shifted from a hazard-focused approach to the broader risk-based perspective covering both physical and societal processes (e.g. Merz et al., 2010, 2014a; Bubeck et al., 2016; Thieken et al., 2016). For instance, the EU Flood Directive (2007/60/EC) has been adopted in October 2007 to launch a flood risk assessment and management framework in Europe considering all aspects of flood risk including the impacts on society.



Conceptually, flood risk is defined as the probability of the adverse consequences within a specified time period.
It depends on three components: hazard, exposure and vulnerability (IPCC, 2012; UNISDR, 2013). Following this
definition, flood risk assessment starts with quantifying the hazard. By combining hazard and socio-economic
information, such as land use and asset values, exposure is assessed. Vulnerability is included by adding
information on how flood-affected objects would be damaged. Overall, flood risk assessment attempts to estimate
the characteristics, e.g. inundation depth and flood extent, of a range of potential flood events, the exceedance
probabilities of these events and their consequences (e.g. Winsemius et al., 2013; de Moel et al., 2015). The results
of flood risk assessments are often presented in maps, which exist in many different forms depending on their
purpose (Merz et al., 2007; de Moel et al., 2009). Flood hazard maps contain flood characteristics, e.g. inundation
extent, water depth, for given return periods. Flood risk maps additionally consider the adverse consequences, e.g.
economic damage, number of affected people.

Flood mapping is typically based on a number of spatially uniform (or homogeneous) scenarios with given return
periods (e.g. Rhine Atlas (ICPR, 2015)). The scenario with T-year return period is composed of all flooded areas
within the study area, whereas each location shows the T-year flood. Hence, the T-year flood map is produced by
piecing together or mosaicking estimates of the local T-year flood based on extreme value statistics at individual
gauges, assuming complete dependence between different locations. Based on this assumption, Ward et al. (2013)
and Winsemius et al. (2013, 2015) estimated flood hazard and risk at the global scale, assuming homogeneous
return period scenarios within regions. At the European scale, flood hazard and risk were assessed based on
spatially homogeneous scenarios by Feyen et al. (2012), Rojas et al. (2013) Alfieri et al. (2014) and Bubeck et al.
(2019). At the national scale, Dumas et al. (2013) investigated future flood risk in France, and Hall et al. (2005)
assessed current and future flood risk in England and Wales by assuming uniform return periods for all flooded
areas. Similarly, te Linde et al. (2011) estimated current and future flood risk along the river Rhine. Real flood
events are, however, spatially heterogeneous as the flood generation depends on a range of processes in the
atmosphere, catchment and river network, which vary strongly in space (e.g. Nied et al., 2017; Vorogushyn et al.,
2018). The analysis of historical floods shows that return periods of peak discharges are typically very
heterogeneous for a given event (Lammersen et al., 2002; Uhlemann et al., 2010; Merz et al., 2014b; Schröter et
al., 2015).

Some studies consider the spatial variability of return periods of floods. One approach applies multivariate
distribution functions to represent the dependence between flood peaks at multiple sites (e.g. Keef et al., 2009;
Lamb et al., 2010; Ghizzoni et al., 2012; Thieken et al., 2015; Quinn et al., 2019). Based on a stochastic dependence
model, spatially heterogeneous scenarios are generated and used for the risk assessment. This approach provides,
however, only flood peaks, whereas the transformation of peaks into inundation areas requires event hydrographs.
Hence, synthetic hydrographs are associated with the peaks, which is an additional source of uncertainties and
errors (Grimaldi et al., 2013). These hydrographs are spatially inconsistent, i.e. not mass conservative, (though
peaks are spatially consistent) and can be used for hydraulic calculations only for a limited river stretch. Another
approach, proposed by Alfieri et al. (2015, 2016, 2017), combines inundation maps and resulting risk for
heterogeneous return periods piece-wise by interpolating between previously derived homogeneous return period
maps. The spatially variable discharges are derived from a hydrological model driven by observations or climate
models. This approach considers spatial dependence, but still suffers from inconsistencies of inundation maps



mosaicked piecewise. Further, an event-based simulation approach, where stochastic precipitation events are generated as input to a hydrological model, has been used (e.g. Rodda, 2001; Jankowfsky et al., 2016). The hydrological model simulates spatially dependent discharge hydrographs, which are then used by the hydrodynamic model to map inundated areas. A disadvantage of this approach is that the return period of discharge is assumed to be equal to the return period of precipitation; an assumption that does not necessarily hold. An alternative approach is a continuous hydrological-hydrodynamic simulation driven by long-term synthetic climate time series (e.g. Falter et al., 2015; Grimaldi et al., 2013). This approach is computationally expensive, however, it has a number of advantages as discussed by Falter et al. (2015). Within the context of this paper, its most relevant advantage is that spatially consistent flood events can be modelled by considering the spatial dependence of the precipitation and of the flood generation processes in the catchment and river network.

According to our literature review, only a few studies consider spatial dependence when assessing flood risk. The large majority assumes spatially homogeneous scenarios. This assumption is also the basis for flood hazard mapping, for instance, in Europe (de Moel et al., 2009), in Iowa in the US (Gilles et al., 2012), in Bangladesh (Tingsanchali and Karim, 2005) and in Honduras (Mastin, 2002). The assumption of complete dependence is appropriate for local risk estimates, but it may bias the risk estimates for larger areas. The purpose of our paper is to investigate this bias. To understand the effect of spatial dependency on risk estimates, we compare three assumptions of spatial dependence: (i) spatially dependent flood events with homogeneous return periods (complete dependence), (ii) spatially dependent events with heterogeneous return periods (modelled dependence), and (iii) spatially independent events with heterogeneous, i.e. randomly selected, return periods (complete independence). We explore the variation of the dependence effect with spatial scale and flood magnitude.

To the best of our knowledge, our study is the first in-depth analysis of this bias at the scale of a large river basin. Lamb et al. (2010) and Wyncoll and Gouldby (2015) compared risk estimates for these three assumptions for smaller regions in UK only (Leeds, York: around 12,000 km², northeast England: around 15,000 km² in the former; Eden catchment: approximately 2,400 km² in the latter); the effect of spatial dependence over large regions has not been studied. Further, they statistically generated spatially dependent peak flows and did not consider the spatial dependence of the flood generating processes as it is possible with the continuous simulation approach of Falter et al. (2015). Jongman et al. (2014) assessed the effect of spatial dependence of flood peaks on flood damage in Europe but considered only modelled dependence versus full independence. They did not analyse the widespread assumption of homogeneous return periods.

To realistically represent the spatial dependence of the different flood processes, we use the derived flood risk analysis (DFRA) based on continuous spatially consistent modelling of the entire flood process chain (Falter et al., 2015). The model chain includes all processes from the precipitation through the catchment and river system to the damage mechanisms. The effect of spatial dependence is investigated for the Elbe catchment in Germany.

This paper is organized in six sections. Section 2 introduces the study area. Section 3 describes the model chain and how the risk estimates are obtained for the three dependence assumptions. Section 4 illustrates the risk estimation results under three spatial dependence assumptions. Further, we discuss these results in Section 5 and draw conclusions in Section 6.


## 2. Study Area: The Elbe Catchment

The river Elbe is located in Central Europe with a length of 1,094 km and total catchment area of 148,268 km$^2$. It can be subdivided into three parts: the Upper Elbe, the Middle Elbe and the Lower Elbe. The Upper Elbe mainly belongs to the Czech Republic and is dominated by mountains. In Germany, the Upper Elbe reaches the north

120   German lowlands at Castle Hirschstein followed by the Middle Elbe reaching the weir Geesthacht. The Lower Elbe starts downstream of Geesthacht and forms the Elbe estuary. Approximately two-thirds of the catchment belong to Germany with the main tributaries Black Elster, Mulde, Saale and Havel (Figure 1). In the present study, the analyses are presented for 29 sub-basins located within Germany. The complete Elbe catchment receives 628 mm precipitation per year, and the characteristic runoff regime is the rain-snow type (Nied et al., 2017).

125   Floods occur mainly in winter and spring, often as snowmelt or rain-on-snow floods. However, the largest floods tend to occur in summer. Heavy precipitation events associated with Vb cyclones have caused disastrous floods, such as the events in August 2002 and June 2013. The 2002 (8.9 billion € damage) and 2013 flood events (5.2 billion €) were the most severe flood events in the Elbe river basin in Germany for the last few decades (IKSE, 2015). Besides, the river basin was affected by smaller floods in 2006, 2010 and 2011.
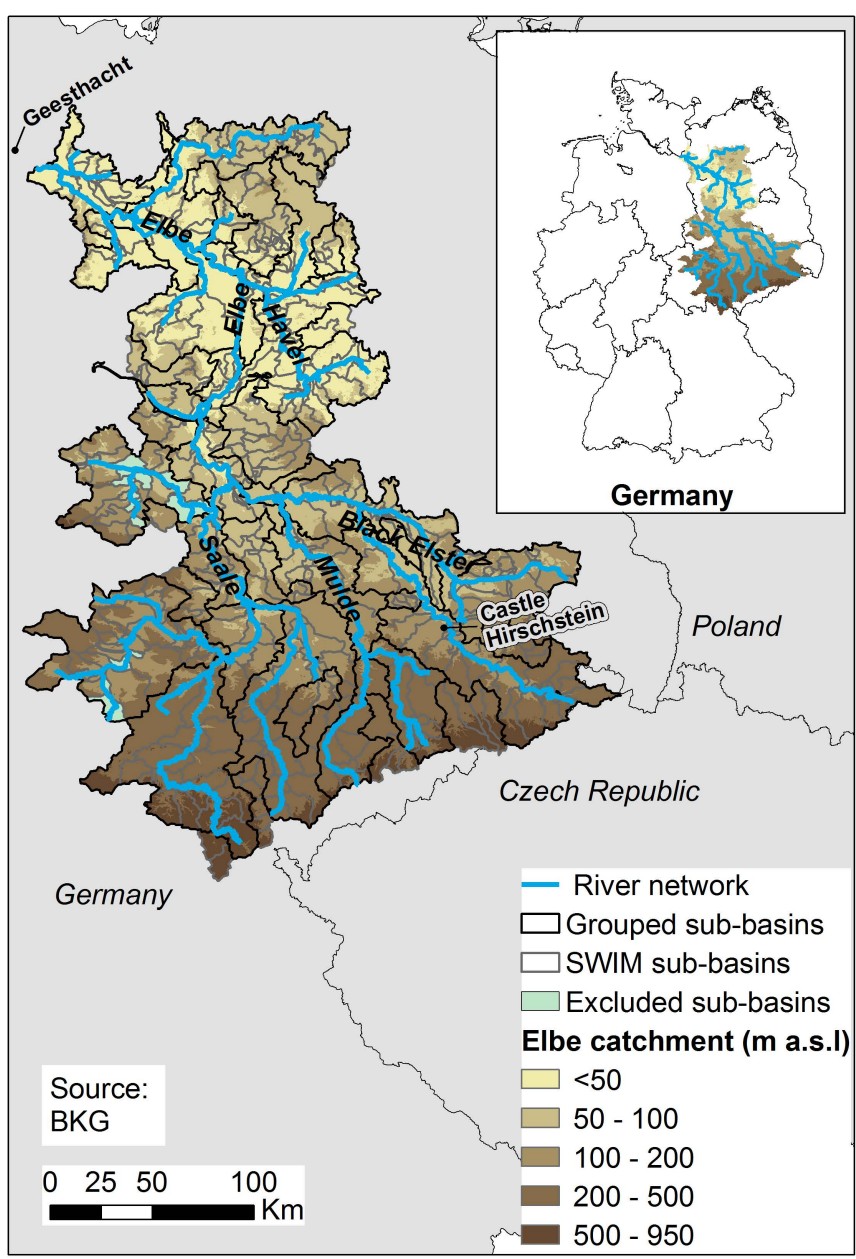

130

**Figure 1.** Study area in the Elbe catchment including the main tributaries and sub-basins. The inset shows the location of the catchment within Germany, data sources of figure: BKG (2012),

### 3. Methods

### 3.1. Regional Flood Model (RFM) for Germany

The Regional Flood Model (RFM) has been developed for large-scale flood risk assessments, i.e. for areas of up to several 100,000 km$^2$. RFM is composed of a weather generator, rainfall-runoff model, 1-D channel routing model, 2-D hinterland inundation model, and flood damage estimation model. The output from one model is used as input for the next model (Figure 2). All processes along the entire flood risk chain are continuously simulated in a distributed manner. Consequently, spatially coherent precipitation patterns and flood pre-conditions of the catchment, including their influence on discharge peaks, water levels, inundation areas and damages are considered.

In this study, RFM is run for time series of 10,000 years (100 realization of 100 years) on a daily time step. Synthetic meteorological time series at multiple sites are provided by a multi-variate weather generator. Further, continuous flood hydrographs at the sub-basin scale are calculated by a hydrological model, where antecedent catchment conditions are implicitly considered. The flow hydrographs are used as a boundary condition for the calculation of water levels in the river channels and inundation depths with a coupled 1D-2D hydrodynamic model considering levee overtopping. Finally, damage time series using a multi-variate flood loss estimation model for residential buildings are simulated. In this way, spatially consistent time series of flood damages at the sub-basin scale are derived. The model components are briefly described in the following. Details about RFM and calibration/validation results of the model components can be found in Falter et al. (2015; 2016) and Metin et al. (2018).

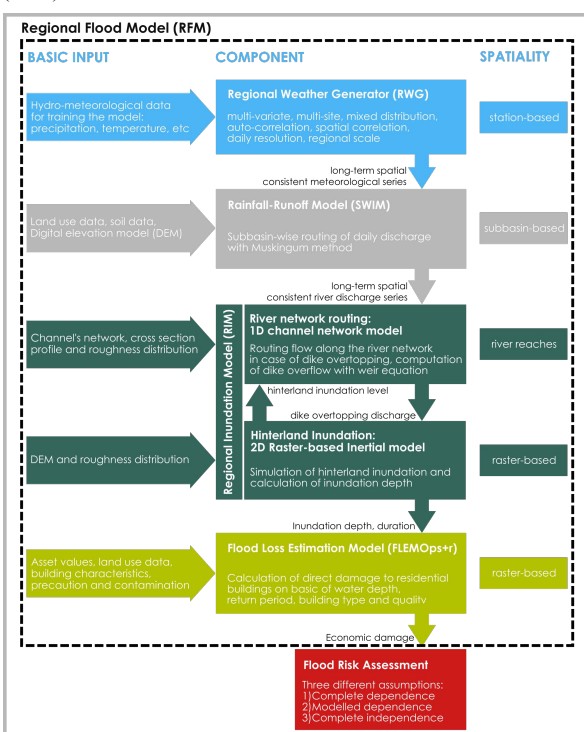

**Figure 2.** Workflow for the derived flood risk assessment (DFRA) with the Regional Flood Model (RFM).


### 3.1.1. Regional Weather Generator RWG

The regional weather generator (RWG), proposed by Hundecha et al. (2009) and further developed by Hundecha and Merz (2012), generates synthetic weather at the regional scale. This multisite, multivariate auto-regressive model generates daily time series of meteorological variables taking into account the spatial correlation structure. First, it generates daily precipitation series using the mixed Gamma-Pareto distribution fitted to the observed data. Further, the model generates daily maximum, minimum, and mean temperature and solar radiation using Gaussian distributions conditioned on precipitation. RWG was setup for the area covering the entire Elbe, Rhine, Danube and Ems rivers using the observed climate data at 528 climate stations between the year 1951 and 2003 and was shown to capture precipitation extremes well (Hundecha et al., 2009).

### 3.1.2. Rainfall-Runoff Model SWIM

Discharge time series on a daily basis are derived with the semi-distributed hydrological model SWIM (Krysanova et al., 1998). The model has a three-level structure of spatial disaggregation: basin, sub-basins and hydrotopes. A hydrotope is a set of disengaged elementary units within the sub-basins, which are homogeneous in terms of land use and soil type. The hydrological processes, such as evapotranspiration, infiltration and snow melt, and different types of runoff are computed at the hydrotope level. The outputs from hydrotopes are integrated (area-weighted average) for each sub-basin. The runoff is routed by the Muskingum routing method between individual sub-basins and is aggregated at the basin scale.

The Elbe catchment was discretized into 2,268 sub-basins in the watershed delineation of the SWIM model (SWIM sub-basins). A detailed soil map (BÜK 1000 N2.3, generated by the Bundesanstalt für Geowissenschaften und Rohstoffe, Hannover) and land use data (the CORINE land cover map) were used. The model was calibrated using observed daily climate data for the period 1981-1989. It was validated with observed discharge data on 20 gauging stations in the Elbe catchment for 1951-2003 (Falter et al., 2015; 2016; Metin et al., 2018). While discharge is simulated well in most parts of the Elbe catchment, peak flows are over- and underestimated in the range of ± 5% throughout most of the catchment (Falter et al., 2016). Discharge is mainly underestimated in Mulde and Black Elster and is overestimated in Saale. The model shows a poor performance for a few small SWIM sub-basins in the upstream part of the Saale catchment likely due to not capturing reservoir effects. These areas are excluded in the present study (Figure 1).

### 3.1.3. Regional Inundation Model RIM

The regional inundation model (RIM) simulates the water level along the river network and hinterland inundation depths. RIM consists of two-way coupled models: a 1D hydrodynamic channel routing model based on the diffusive wave equation and a raster-based 2D hydrodynamic inundation model based on the inertia formulation (Metin et al., 2018). The overtopping flow is calculated by the 1D model and is used as boundary condition for the 2D model, which is back-coupled to the 1D model. The river geometry is described by simplified cross-sections which include the overbank river geometry and dike crest elevation derived from the 10 m DEM provided by the Federal Agency for Cartography and Geodesy in Germany (BKG). Whenever the water level overtops the dike crest elevation, the overtopping flow is computed using the broad-crested weir equation.


The river profiles were manually extracted perpendicular to the flow direction every 500 m. Due to low resolution of the DEM 10 in relation to the dike geometry, the derived dike heights tend to be lower than in reality. Hence, a minimum dike height of 1.8 m was used for the river Elbe. A constant Manning's roughness of 0.03 was assumed in the river network. For the 2D raster-based model, DEM 10 m was resampled to 100 m computational grid found

to well represent the inundation characteristics with suitable computation time (Falter et al., 2013).

The 1D hydrodynamic channel routing model was validated with observed data for 1951–2003 at eight gauging stations in the Elbe catchment (Falter et al., 2015; 2016). The performance of the 1D model is acceptable even though there is a tendency to underestimate observed peak flows exceeding the bankfull depth. The simulated inundation areas were compared to the extreme flood in August 2002, the only event for which inundation depth

and extent are available. Although the model tends to underestimate inundation extents, since it neglects dike breaches, it provides plausible inundation patterns.

### 3.1.4. Flood Loss Estimation Model FLEMOps+r

The direct economic damage to residential buildings is estimated by the Flood Loss Estimation Model for the private sector (FLEMOps+r). The model considers five inundation depth classes, two building quality classes (high

quality or medium/low quality), three building types (single-family, semi-detached/detached or multifamily houses) and three return period classes to estimate damage (Elmer et al., 2012). The model provides the damage ratio which is multiplied with the asset values of the inundated residential buildings to obtain the monetary damage.

Besides inundation depths and return periods, the model requires spatially detailed information on building qualities, building types and asset values. The mean building quality and cluster of building type composition was

estimated on the municipal level on basis of Infas Geodaten GmbH (2009). The asset values were determined considering the standard construction costs (BMVBW, 2005) and were spatially disaggregated using the digital basic landscape model ATKIS Basis DLM (BKG 2009). Municipal asset data were disaggregated by means of a dasymetric mapping approach (Wünsch et al., 2009). The damage was estimated according to output from the hydrodynamic model on a raster level by calculating the damage ratio according to the inundation depth and return

period in the corresponding cell and the underlying information for building types and qualities per municipality (Thieken et al., 2008).

The model was validated on the micro- and meso-scale on basis of empirical damage data of the August 2002 flood in the State of Saxony in Germany (Elmer et al., 2010; Falter et al., 2015).

### 3.2. Flood Risk Assessment for Different Dependence Assumptions

We compute flood risk for three spatial dependence assumptions (Figure 3): (1) complete dependence or homogeneous return periods across the river basin, (2) modelled dependence or heterogeneous return periods, and (3) complete independence, where flood peaks and associated return periods are randomly sampled. In scenarios (1) and (3) the discharges, inundation areas and damages are spatially inconsistent, i.e. they are mosaicked from the continuous simulations by selecting events and damages for corresponding return periods. The spatial variation

of damages within the catchment depends on the spatio-temporal patterns of meteorological, hydrological and hydraulic processes. For instance, the flood damage downstream of the confluence of two tributaries depends on the superposition of the flood waves from these tributaries. The damage results of the modelled dependence should



lie between the results of the two other assumptions as they span the whole range from complete dependence to complete independence. Further, the modelled dependence results should be similar to those of the complete 230 dependence for small areas, and should move towards complete independence as the spatial scale becomes large.

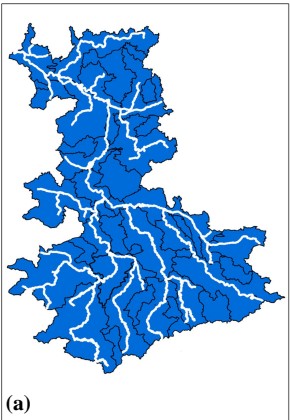 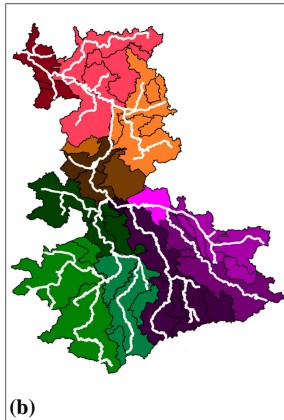 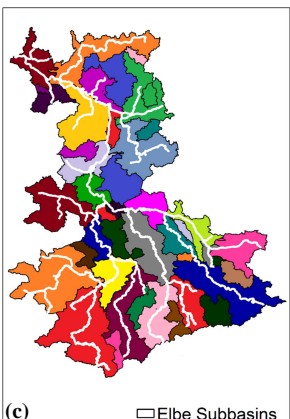

**Figure 3.** Conceptual representation of the three assumptions on spatial dependence: (a) complete dependence; (b) modelled dependence; and (c) complete independence. Return periods of damage are color-coded at the sub-basin level.

We characterize flood risk by the probability of damage (risk curve) and by the expected annual damage (EAD) computed as the integral of the risk curve. Damage values are calculated at the 100 m grid resolution for individual inundation events of the 10,000-year continuous flood simulation with RFM. An event requires that flood defences are overtopped at least at one location and affects residential assets, i.e. a non-zero damage occurs. If anywhere in the entire catchment overtopping occurs after at least 10 days of non-overtopping, this is defined as the start of a 240 new event. Empirical return periods for damages aggregated for specific spatial units (e.g. sub-catchments) are determined using Weibull plotting positions. Damage at the level of the sub-basins (SWIM sub-basins) is then aggregated to larger spatial units (e.g. aggregation of sub-basins or the entire catchment) for individual flood events. These pairs, i.e. damage and associated return period, are used to construct risk curves and to calculate EAD (Falter et al., 2015).

Under the assumption of complete dependence, all sub-basins within the considered spatial unit, e.g. the entire river basin, are assumed to experience a T-year flood damage at the same time. Hence, the T-year flood damage is calculated by aggregating the T-year damage values of all sub-basins estimated from individual (not necessarily concurrent) events. In the following, we refer to a T-year flood event as an event resulting in the T-year damage.

Under the modelled dependence assumption, damages are aggregated for individual flood events across the 250 considered spatial unit, and return periods of aggregated damages are derived directly for this spatial unit. This approach aims to represent the true spatial and temporal dependencies of real world flood situations. For example, for a T-year flood loss over the entire catchment, the return periods of damages in individual sub-basins are





expected to be different from sub-basin to sub-basin. Furthermore, these return periods are expected to show a certain spatial pattern dictated by the spatial correlation of the flood generation processes.

Under the assumption of complete independence, the spatial correlations between damages of different sub-basins are destroyed. Damages of individual flood events are shuffled at the SWIM sub-basin level and aggregated for the considered spatial units. Return periods of these aggregated damages are determined for the spatial unit considered. As the aggregated damage and the risk curve depend on the specific realization of the shuffling, this procedure is repeated 1,000 times. From this sample, the median is used to construct the risk curve and additionally
the 95% confidence range is computed.

The risk curves and EAD are derived at the grouped sub-basin level (29 sub-basins in total, see Fig.1), as a higher resolution would lead to many instances where the number of damaging floods would be too low to construct meaningful empirical risk curves.

**4. Results**

**4.1. Damage Estimations under three Dependence Assumptions for the Entire Catchment**

The aggregated economic damages to residential buildings for the Elbe catchment and their corresponding return periods are illustrated in Figure 4 for the three dependence assumptions. While the economic damage of the 1,000-year event is estimated at around €620 million under the assumption of complete dependence, it is around €360 million for the modelled dependence scenario (70% overestimation under the assumption of complete
dependence). A strong overestimation is also given for smaller return periods down to approximately 150 years. Moreover, the assumption of complete independence may underestimate damage by 50%. The extreme assumption of complete independence represents the lower limit for large return periods. For smaller return periods, however, we see the opposite effect. The damage is underestimated under the assumption of complete dependence for events with return periods smaller than 87 years.

The point where the risk curves of modelled dependence and complete dependence intersect is called the 'intersection point' in the following. For return periods up to this intersection point, the complete dependence assumption underestimates the damage compared to modelled dependence; all sub-basins show either no or small damages as the flood peaks are mostly below the flood defences. However, for the assumption of modelled dependence, the return periods vary and a small to medium return period event at the scale of the entire Elbe
catchment may be composed of many sub-basins without any damage but a few sub-basins with large damage, because in these sub-basins the flood defences are overtopped. A similar explanation holds for the situation beyond the intersection point: The complete dependence assumption leads to events where all sub-basins tend to show large damages as flood defences are overtopped everywhere. In contrast, under the modelled dependence assumption many sub-basins show large damages as defences are overtopped, however there are also sub-basins
without damage as consequence of spatial variability.


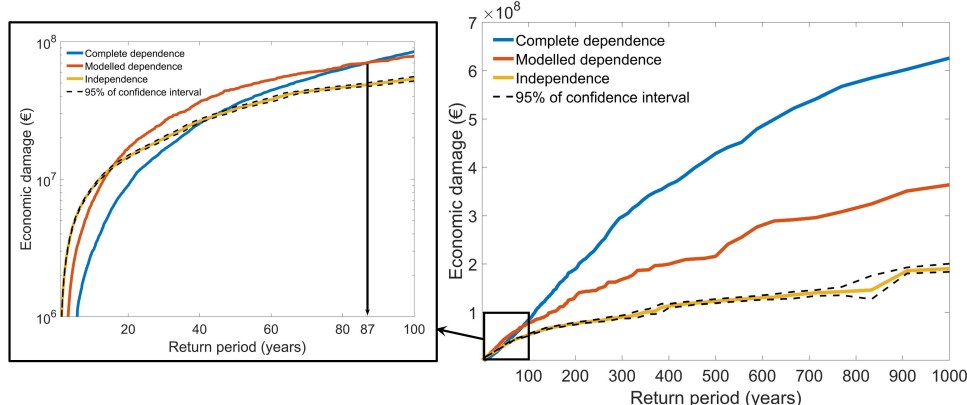

**Figure 4.** Risk curves for the Elbe catchment for three dependence assumptions (complete dependence, complete independence and modelled dependence). The left panel zooms in the risk curves up to the 100-year return period of damage.

The underestimation (overestimation) for small (large) return periods under the complete dependence assumption is a consequence of the strong link between the damages of the different sub-basins. For a better understanding, Figure 5 illustrates the spatial distribution of damages at the sub-basin level for the three dependence assumptions that lead to the 20- and 200-year event at the catchment scale, respectively. For the 20-year event, under the complete dependence assumption, all sub-basins show either no damage or small to medium damage, leading to

comparatively small damage at the scale of the entire basin (Figure 5a). The 20-year event for the modelled dependence assumption consists mainly of sub-basins without any damage, but due to dike overtopping single sub-basins may experience a high damage. These sub-basins are clustered, in this case in the northwest of the Elbe catchment, illustrating the effect of spatial dependence. In contrast, the damages are not clustered under the complete dependence and independence assumptions. For the 200-year event (Figure 5b), almost all sub-basins

indicate large damage under the complete dependence assumptions, resulting in the overestimation under complete dependence assumption for the entire catchment.


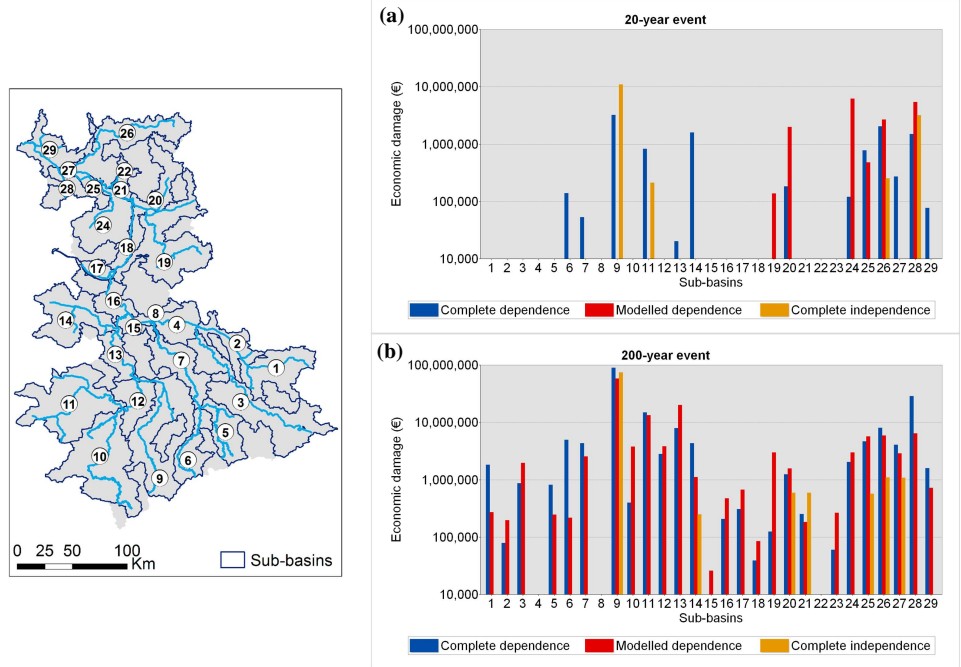

**Figure 5.** Distribution of damages at the sub-basin level for (a) the 20-year event and (b) 200-year event for three dependence assumptions.

**4.2. Variation of Damage Estimations with Spatial Scale under three Dependence Assumptions**

To understand how the risk estimates for the three dependence assumptions vary with spatial scale, the risk curves for aggregations of sub-basins from upstream to the entire catchment are given in Figure 6. As a general rule, smaller regions should be characterized by stronger spatial dependence of damage. This should lead to (1) an increasing difference between the risk curves of the three dependence assumptions with increasing scale, and (2) 310 a shift of the modelled dependence risk curve from the complete dependence towards independence with increasing scale. Both effects are seen in Figure 6.

The intersection point shifts from return periods of a few hundred years for smaller aggregation areas, i.e. sub-basins 1 to 8 (up to 11,800 km², upper panel in Figure 6), to approximately 90 years for the larger areas. The intersection point is mainly affected by the threshold where damage occurs, i.e. by the flood protection or elevated 315 banks. This strong shift in the intersection point is however not a consequence of very different flood defence standards in the up- and downstream parts of the Elbe catchment, but rather results from data and modelling errors. In particular, the small scale variability of precipitation extremes appeared to be insufficiently well captured by the weather generator in some sub-basins due to varying station density used for paramaterization. Sub-basins 1 to 8 (Mulde and Black Elster rivers) experience very small damage even for high return periods, while the opposite 320 is true for sub-basins 9 to 14 (Saale river). This is explained by the underestimation of damage for the Mulde and Black Elster rivers and overestimation for the Saale River.


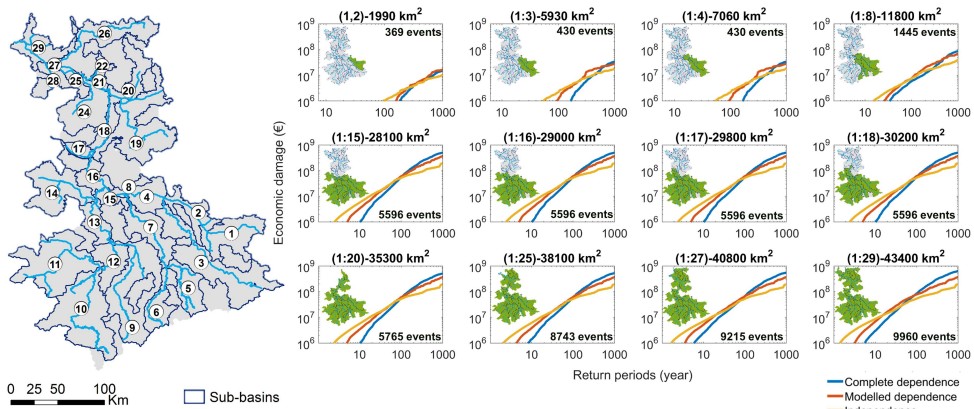

**Figure 6.** Sub-basins in the Elbe catchment (left) and risk curves of aggregations of sub-basins (right) under complete dependence, modelled dependence and independence. The aggregated sub-basins are ordered along increasing scale and are denoted by the green colour within each risk curve and the colon (:) between start and end sub-basin numbers.

### 4.3. Errors in Expected Annual Damage (EAD) and in 200-year Damage under 'false' Assumptions of Spatial Dependence

Besides the risk curve, expected annual damage (EAD) and the damage for a T-year return period are important risk measures. We assess here the 200-year return period damage which is particularly important for the insurance sector. Their percentage error under the complete dependence and independence assumptions, compared to the modelled dependence assumption, is given in Figure 7 for the entire Elbe catchment. The false assumptions about spatial dependence do not impact the EAD estimation. EAD is the sum of 29 random variables, i.e. the damages for the 29 sub-basins. As the mean value of a sum of random variables is not influenced by the correlation between the variables, the spatial correlation can be neglected when one is only interested in EAD. However, correlation influences the variance of a sum of random variables. Hence, for other values, such as the 200-year event, it is crucial to include the 'true' spatial dependence pattern. In our case, the damage for the 200-year event is overestimated (underestimated) under complete dependence (independence) by around 40%.



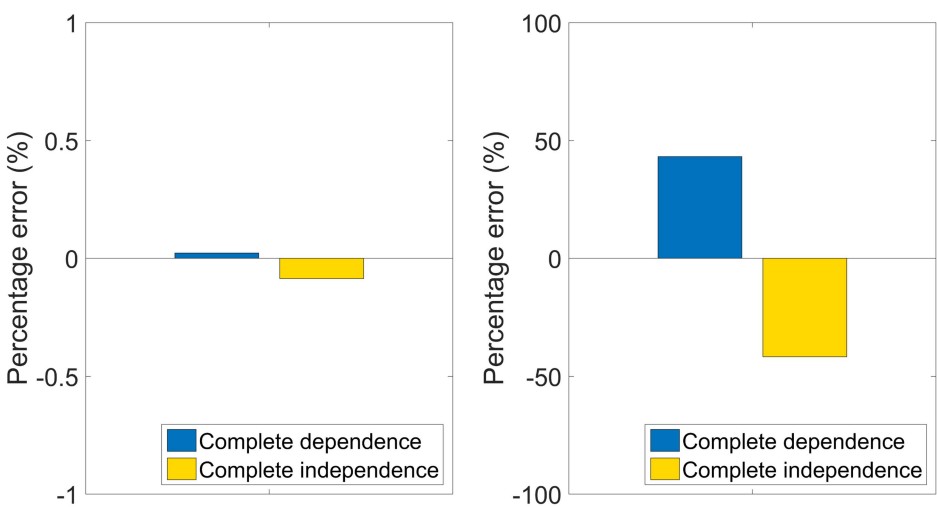


**Figure 7.** Percentage error in expected annual damage (EAD) (left) and in economic damage for the 200-year event (right) under the assumptions of complete dependence and complete independence for the entire Elbe catchment.

**5. Discussion**

This study investigates the effects of spatial dependence of flood generation processes on risk estimates. It compares the 'true' dependence scenario to the two endpoints, i.e. complete dependence and complete independence. It is shown that the assumption of complete spatial dependence, which is often used in risk assessments, leads to under- and overestimation of flood risk for small and large return periods, respectively.

Although several papers have suspected that the complete dependence assumption may bias risk estimates, this
bias has been investigated by the two studies of Lamb et al. (2010) and Wyncoll and Gouldby (2015) only. As these studies are limited to small and medium study areas up to 15,000 km², our study is the first investigation for a large-scale river basin. In addition, our study uses a more elaborate setup, as the spatial dependence of all processes along the flood risk chain, from the precipitation to the damage, is included. The larger study area allows us to investigate how the differences in risk estimates change with increasing scale. The modelled dependence
estimate tends to be similar to the complete dependence scenario for smaller areas and to shift towards the independence scenario when the scale is increased. However, this shift is not very prominent. We assume that the variety of processes that are involved in the generation of damage blurs a clear signal when going from smaller to larger scales. The space-time dynamics of flood damage events is not only influenced by the space-time dynamics of the triggering rainfall event and antecedent catchment conditions, but also by the topology of the river network,
flood wave superposition, structural flood defences and the spatial distribution of the asset values and their vulnerability. More work is needed to better understand how the spatial dependencies of different processes along the risk process chain influence the mismatch between modelled and complete dependence. If simple rules could be derived, they could be used to decide whether the spatial dependence of the damage generating processes need to be taken into account or whether a simplified analysis neglecting spatial dependence would suffice.





We are not aware of any study which discussed the intersection point between modelled and complete dependence. We show that the overestimation of risk by the complete dependence assumption that has been reported by Lamb et al. (2010) and Wyncoll and Gouldby (2015) applies to large return periods only. For small return periods the complete dependence assumption underestimates risk. This behaviour, and the location of the intersection point, are mainly affected by the damage threshold controlled by the flood defence level or elevated banks.

Although each model in RFM has some limitations, RFM represents well the spatial patterns of the different flood generation processes (Falter et al., 2016, Metin et al., 2018). For this study, the model limitations are not seen as major concerns because the different assumptions on spatial dependence are investigated by using the same model. The largest limitation related to the current study is probably that dikes can only be overtopped but do not breach. Hence, the number of inundation events and damages may be underestimated. This could affect the intersection

point, i.e. the point where the underestimation of the complete dependence turns into overestimation. Including dike breaches in the model might shift the intersection point to smaller return periods.

   As expected from statistical reasoning, our study confirms that the expected annual damage (EAD) is not biased by false assumptions on spatial dependence. If one is only interested in EAD, spatial dependence can be neglected which drastically simplifies the analysis. However, EAD is a rather limited indicator of risk as discussed, for

instance, by Merz et al. (2009). Further, specific purposes demand assessments of certain risk scenarios for which spatial dependence is crucial. According to Article 101 of the European Solvency II Directive, insurance companies are required to prove that they can cover at least damage events with a return period of 200 years (EC, 2009). The spatial dependence in damage is also highly relevant for disaster management or large-scale, strategic flood planning. It is important, for instance, to understand what disaster management resources are needed for

large-scale floods.

### 6. Conclusions

   This paper analysed the impact of spatial dependence in flood damage generation on risk estimates for the large-scale Elbe River basin in Germany. The 'true' spatial dependence was simulated with the continuous flood risk modelling approach proposed by Falter et al. (2015), where all processes, including their spatial dependence, from

the flood triggering rainfall to the damage processes are considered. The bias between the widespread, but false assumption of complete dependence and the modelled dependence was investigated as function of spatial scale.

   Our results show that for extreme events the economic damage can be strongly overestimated when homogeneous return periods are assumed across the catchment. For the Elbe river basin, damage is overestimated by about 40% for the 200-year event and by almost 100% for the 500-year event. On the other hand, for events with small to

medium return periods, the complete dependence assumption underestimates damage. The intersection point where the underestimation turns into an overestimation depends mainly on the damage threshold, i.e. on the flood defence level in protected areas.

   The spatial scale, for which a risk estimate is sought, decides whether the modelled dependence assumption is closer to complete dependence or independence, respectively. The modelled dependence risk curve is closer to

complete dependence for the upstream areas comprising the Mulde and Black Elster rivers; with increasing scale it shifts towards the independent case. Consequently, the overestimation under the complete dependence


assumption increases with larger areas. As a general rule, the true dependence might be approximated by the complete dependence assumption for smaller regions, whereas for larger regions the independence assumption might serve as an approximation in a rough analysis when including the spatial dependence seems to costly or 405 demanding. However, our study does not allow to specify in a generic way at which scales which assumption might serve as approximation. More systematic analyses are necessary to answer this question.

If one is only interested in the expected annual damage (EAD), then false assumptions on spatial dependence do not bias its estimate. Although EAD is an important risk indicator, for example for cost-benefit analyses of flood mitigation or in the insurance sector, we strongly advocate to consider the complete risk curve as it gives a much 410 richer perspective on the risk and the effects of mitigation measures. Hence, we conclude that it is of highest relevance to realistically represent the spatial dependence of flood damage for large-scale risk estimates.

**Data availability.** The data used in this paper are not publicly accessible; however, the authors can be contacted by email (admetin@gfz-potsdam.de, dung@gfz-potsdam.de, kai.schroeter@gfz-potsdam.de) for help in acquiring 415 such data.

**Author contributions.** BM conceived and supervised the study. BM, ADM, NVD, and SV developed the concept. NVD, KS, and ADM performed simulations. ADM analysed the results. ADM prepared the paper with contributions from all the co-authors. All authors made a substantial contribution to the interpretation of results and provided important ideas to further improve the study.

**Competing interests.** The authors declare that they have no conflict of interest.

**Acknowledgements.** We gratefully acknowledge funding from the European Union's Horizon 2020 research and innovation programme under the Marie Skłodowska-Curie grant agreement No 676027 and from the German Research Foundation (DFG Research Group FOR 2416 SPATE). We thank the Federal Agency for Cartography and Geodesy in Germany (BKG) for providing DEM data.

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
