# Peer review of "The role of spatial dependence for large-scale flood risk estimation"

_Natural Hazards and Earth System Sciences, 2019_

## Referee Comment (RC1) · Anonymous Referee #1 · 10 Jan 2020

The authors present a modelling exercise where they compare and analyse flood damages under three different assumptions regarding the spatial distribution of return periods of flood peaks throughout the catchment. They apply a complex modelling chain that goes from synthetic weather generation to flood damage estimates. The chain is composed of a weather generation algorithm, a rainfall-runoff model, a 1-D flood propagation model, a 2-D spatial inundation model and a damage estimation model. Their emphasis is on the comparison of the risk curves obtained throughout the catchment for different assumptions of spatial dependence of flood return periods, taking as reference case the option of modelled dependence. The authors present results of their analyses for the entire Elbe catchment and for its division in 29 sub-basins. They infer conclusions on the over estimation or underestimation of flood damages with respect

to the reference case.

The topic is relevant for the audience of NHESS, the objectives are clearly identified, the methodology for the analysis is adequate and the conclusions are relevant and correctly supported by the results and discussion. The analysis clearly shows the striking differences between the options of independence and complete dependence with respect to the reference condition of modelled dependence, and therefore the objective of the paper is clearly achieved. Therefore, I believe the manuscript deserves publication in NHESS.

SPECIFIC COMMENTS The authors are addressing a formidable task. They are reporting results obtained with a complex modelling chain that probably took years of work under the constraints imposed by the length of a research paper. It is only natural that some parts of their work or the models used have necessarily been left unexplained. I am suggesting a few points where I believe the reader would benefit from some additional details, such as the following:

a) On page 7, lines 161-163, the authors report that the weather generator was calibrated for the region of the Elbe basin with observed data and "was shown to capture precipitation extremes well". I think the entire analysis is dependent on the quality of the time series produced by the weather generator, particularly regarding spatial correlation and seasonality of extremes, which are very challenging. I think the paper would benefit from a more elaborate discussion of the spatial dependence of the extremes produced by the weather generator as compared to observations.

b) The topology of the model should be explained better. On page 7, line 186, the authors state that overtopping flow is calculated from the 1-D diffusive wave model. On line 170, they say that the runoff is routed by the Muskingum method and aggregated at the basin scale. This leads me to think that the possibility of overtopping is not contemplated in the reaches modelled by Muskingum. From Figure 1 I gather that the 29 sub-basins under analysis are actually composed of smaller units, which are the

ones simulated in SWIM, but this is not mentioned in the text. I think the manuscript would benefit form a brief discussion of which rivers are included in the overtopping analysis and which criteria were used to delineate the SWIM basins.

c) The operational definition of return period should also be discussed in detail. I first thought that the return period referred to peak flow and was estimated from the 10,000 year simulation in each location. However, on page 9, line 248, the authors say that they refer to a T-year flood event as resulting in the T-year damage. In addition to peak flow, flood damage is also affected by hydrograph volume, which is very relevant to determine the extent of the inundated area, at least for flash floods. Perhaps the authors might consider a brief discussion of this issue.

TECHNICAL CORRECTION From the formal standpoint, the paper is very well written, correctly organized and adequately illustrated with figures. Comparison of Figures 4 and 6 is handicapped by the fact that the return period is shown in natural scale in Figure 4, while it is shown in logarithmic scale in Figure 6. I would suggest using the same type of scale on both figures, if possible.

---

## Referee Comment (RC2) · ATTILIO CASTELLARIN (Referee) · 30 Jan 2020

MANUSCRIPT ID nhess-2019-393

MANUSCRIPT TILE THE ROLE OF SPATIAL DEPENDENCE FOR LARGE-SCALE FLOOD RISK ESTIMATION

AUTHORS: Metin, A. D., Dung, N. V., Schröter, K., Vorogushyn, S., Guse, B., Kreibich, H., and Merz, B.

GENERAL COMMENT

I congratulate with the authors for their very interesting manuscript on the impact of spatial dependence of hydrologic and hydraulic forcing on flood risk assessment. This

study addresses a very relevant issue in flood risk assessment, that is the value and impact of a commonly assumed working hypothesis, i.e. homogeneous return period of flood events across large geographical areas and regions. This common hypothesis implies a complete dependence (i.e. perfect spatial correlation) of flood events and should be rather regarded as an extreme condition. The hypothesis is contrasted in the study to the opposite extreme situation, namely complete spatial independence of hydrological/hydraulic forcing, and with an intermediate condition that models the real spatial dependence through an ad hoc modelling chain. Impacts of these three hypotheses are quantitatively compared in terms estimated annual damage and risk curve (across different spatial scales) for a large river basin (Elbe river). I found the manuscript to be well structured and sufficiently clear (see comments below) and I only have moderate and minor comments on it.

My recommendation is therefore: return to authors for moderate revisions.

I hope the authors will find my comments useful while revising their manuscript.

With kind regards, Attilio Castellarin

MAJOR CONCERNS

- Levee breaching is neglected

The authors clearly state that they neglected levee-breaching in their analysis. This is a rather strong assumption because it is far from real conditions (once an earthen levee is over-topped, it will most likely breach) and because the formation of a breach has serious implication on the hazard downstream (lower flood peaks and volumes downstream the breach) and in the inundated area (larger outflow volumes if a breach is present relative to the no-breach case). I would suggest deepening the discussion on this main assumption.

- Only direct losses to residential buildings

The authors apply a multiple-variable damage model that considers only direct damages to residential buildings. Given the strong expertise on flood damage modeling of the GFZ research team, I would like to see some discussion on this assumption as well. Based on their previous research activities, could the authors speculate if the same results of the study should apply also to other kind of damages (direct damages to the industrial/agricultural sector, indirect damages)? Are the three hypotheses on spatial dependence really interchangeable if indirect damages (e.g. disruption of services) are considered instead of direct ones?

- Complete spatial dependence / independence

I believe I understood the technicalities for simulating flood damages and assessing flood risk under the hypotheses: (i) complete spatial dependence and (iii) complete independence. Yet, I would say that while the analysis framework is extremely clear for (ii) "modeled dependence" (perhaps also due to the flowchart of Fig. 2), I my opinion the description of the technicalities of how flood risk is modeled under hypotheses (i) and (iii) is not as clear (hydrological simulation is repeated, or streamflow time series are simply resampled?). This is a pity because the manuscript could serve as a blueprint for repeating the same modeling exercise in different areas or at larger scales (continental, global) considering all three spatial dependence typologies. Suggestion: it would be goo to have an illustration on what differs in the flowchart of Fig. 2 when hypotheses (i) and (iii) are considered instead of (ii).

SPECIFIC REMARKS

* L.29 "there were 3062 floods", consider using a rounded figure, floods may occur without impacting people and therefore without being recorded

* Fig. 2, consider enhancing readability (e.g. by using larger fonts)

* Sections 3.1.1 and 3.1.2 consider commenting on the limitations associated with the simulation timescale (daily) relative to result in validation (poorer results in some small catchments)

* L.188, is the continuity of the levee crest ensured using a 10m res. DEM

*L. 193, consider including Manning's roughness units

* Fig.4, I am surprised by the remarkable narrowness of the confidence interval for the fully independent case, is it associated with the size of resampled sets? The authors should comment on this.

* Fig.6, panels could be reported (also) in a standardized form concerning the y-axes, so that the size of each region is neglected, the three curves are more complete in all panes and similarities/dissimilarities between different cases can be better illustrated

---

## Author Comment (AC1) · 26 Feb 2020

**Response to Referee #1**

First of all, we would like to thank the referee for the time and effort put into reviewing the manuscript. This response carefully addresses all the comments. We further attach a change tracked version of the manuscript.

**The authors present a modelling exercise where they compare and analyse flood damages under three different assumptions regarding the spatial distribution of return periods of flood peaks throughout the catchment. They apply a complex modelling chain that goes from synthetic weather generation to flood damage estimates. The chain is composed of a weather generation algorithm, a rainfall-runoff model, a 1-D flood propagation model, a 2-D spatial inundation model and a damage estimation model. Their emphasis is on the comparison of the risk curves obtained throughout the catchment for different assumptions of spatial dependence of flood return periods, taking as reference case the option of modelled dependence. The authors present results of their analyses for the entire Elbe catchment and for its division in 29 sub-basins. They infer conclusions on the over estimation or underestimation of flood damages with respect to the reference case.**

**The topic is relevant for the audience of NHESS, the objectives are clearly identified, the methodology for the analysis is adequate and the conclusions are relevant and correctly supported by the results and discussion. The analysis clearly shows the striking differences between the options of independence and complete dependence with respect to the reference condition of modelled dependence, and therefore the objective of the paper is clearly achieved. Therefore, I believe the manuscript deserves publication in NHESS.**

**SPECIFIC COMMENTS The authors are addressing a formidable task. They are reporting results obtained with a complex modelling chain that probably took years of work under the constraints imposed by the length of a research paper. It is only natural that some parts of their work or the models used have necessarily been left unexplained. I am suggesting a few points where I believe the reader would benefit from some additional details, such as the following:**

We would like to thank the reviewer for his/her positive feedback and kind remarks.

**a) On page 7, lines 161-163, the authors report that the weather generator was calibrated for the region of the Elbe basin with observed data and "was shown to capture precipitation extremes well". I think the entire analysis is dependent on the quality of the time series produced by the weather generator, particularly regarding spatial correlation and seasonality of extremes, which are very challenging. I think the paper would benefit from a more elaborate discussion of the spatial dependence of the extremes produced by the weather generator as compared to observations.**

We thank the referee for this suggestion. We agree that the quality of the time series produced by the weather generator is significant for the entire analysis. In response to this comment, we provide the figure below which compares the simulated (grey circles, each per one simulation realization) and observed (red-filled triangle) daily extreme precipitation (99.9th percentile) at 9 selected stations (out of 528 stations) over the simulation area. It shows that daily precipitation extremes and their seasonality are captured very well by the weather generator at the locations of the individual stations. We currently carry out a spatial analysis of the weather generator performance and together with the at-site validation, this will be a self-consistent study on the weather generator. We do observe that the spatial rainfall performance is more challenging for the weather generator than capturing the local statistics as also known for other generators discussed in the literature (e.g. Serinaldi and Kilsby 2014). It seems that the spatial dependence of very strong rainfalls is somewhat overestimated which will presumably translate into the dependence of discharge peaks. However, we believe that for the purpose of the presented analysis, the overestimation of spatial rainfall dependence is not critical. The results of modelled dependence are located between complete dependence and complete independence for high return periods. With an ideal weather generator, they would be closer to the complete independence. Thus, our estimates for the difference between the assumption of complete dependence and modelled dependence can be regarded as conservative. Hence, the major conclusion challenging the assumption of homogeneous return periods still holds. This will be added to the revised version of the manuscript.

[Figure]

**b) The topology of the model should be explained better. On page 7, line 186, the authors state that overtopping flow is calculated from the 1-D diffusive wave model. On line 170, they say that the runoff is routed by the Muskingum method and aggregated at the basin scale. This leads me to think that the possibility of overtopping is not contemplated in the reaches modelled by Muskingum. From Figure 1 I gather that the 29 sub-basins under analysis are actually composed of smaller units, which are the ones simulated in SWIM, but this is not mentioned in the text. I think the manuscript would benefit form a brief discussion of which rivers are included in the overtopping analysis and which criteria were used to delineate the SWIM basins.**

In this study, regional flood model (RFM) considers SWIM sub-basins in the calculations. Only for a better representation, the risk results are presented for grouped sub-basins (29 sub-basins), which aggregate several original SWIM sub-basins. In SWIM model, the entire catchment area is first subdivided into sub-basins with average area in a range of 10 to 100 km$^2$ and definitely not larger than 100 km$^2$. Additionally, in response to overtopping analysis, the reviewer is right, the overtopping is considered only at the main river network and higher order tributaries that have a drainage area of 600 km$^2$ or more. This river network is explicitly modelled with the 1D-diffusive wave hydrodynamic model. The flood routing in smaller tributaries with drainage area below the above-mentioned threshold is done by the Muskingum routing within the SWIM model. These will be added to the revised version of the manuscript.

**c) The operational definition of return period should also be discussed in detail. I first thought that the return period referred to peak flow and was estimated from the 10,000 year simulation in each location. However, on page 9, line 248, the authors say that they refer to a T-year flood event as resulting in the T-year damage. In addition to peak flow, flood damage is also affected by hydrograph volume, which is very relevant to determine the extent of the inundated area, at least for flash floods. Perhaps the authors might consider a brief discussion of this issue.**

We thank the reviewer for pointing this out. Flood damage depends not only on the flood peak but on the hydrograph shape, floodplain hydraulics (e.g. dike overtopping and inundation patterns), exposure and vulnerability of affected elements. 10,000-year simulation of the risk chain enables us a large sample of damages.

From this sample, we derive an empirical frequency distribution for the probability of damage which is then used in the estimation of flood risk. Hence, we refer to a T-year flood as a flood resulting in the T-year damage in this study. We denote here the term "T-year flood" in a different way based on damage return period as compared to the traditional way based on the peak flow return period. This will be added to the revised version.

**TECHNICAL CORRECTION From the formal standpoint, the paper is very well written, correctly organized and adequately illustrated with figures. Comparison of Figures 4 and 6 is handicapped by the fact that the return period is shown in natural scale in Figure 4, while it is shown in logarithmic scale in Figure 6. I would suggest using the same type of scale on both figures, if possible.**

We thank the reviewer for this comment. In fact, in Figure 6, bottom-right panel, we illustrate the risk curves for entire catchment (1:29) on a logarithmic scale. Therefore, this comparison is still possible and we would prefer not to change these figures. We provide risk curves on a linear scale in Figure 4 for a better understanding of the damage ranges.

**Reference**
Serinaldi F., Kilsby C. G. (2014): Simulating daily rainfall fields over large areas for collective risk Estimation, Journal of Hydrology, 512, 285-302, doi: 10.1016/j.jhydrol.2014.02.043.

---

## Author Comment (AC2) · 26 Feb 2020

**Response to Referee #2**

Dear Attilio Castellarin, we would like to thank you for the time and effort put into reviewing the manuscript. This response carefully addresses all the comments. We further attach a change tracked version of the manuscript from which the changes proposed can be seen.

**GENERAL COMMENT**
**I congratulate with the authors for their very interesting manuscript on the impact of spatial dependence of hydrologic and hydraulic forcing on flood risk assessment. This study addresses a very relevant issue in flood risk assessment, that is the value and impact of a commonly assumed working hypothesis, i.e. homogeneous return period of flood events across large geographical areas and regions. This common hypothesis implies a complete dependence (i.e. perfect spatial correlation) of flood events and should be rather regarded as an extreme condition. The hypothesis is contrasted in the study to the opposite extreme situation, namely complete spatial independence of hydrological/hydraulic forcing, and with an intermediate condition that models the real spatial dependence through an ad hoc modelling chain. Impacts of these three hypotheses are quantitatively compared in terms estimated annual damage and risk curve (across different spatial scales) for a large river basin (Elbe river). I found the manuscript to be well structured and sufficiently clear (see comments below) and I only have moderate and minor comments on it. My recommendation is therefore: return to authors for moderate revisions. I hope the authors will find my comments useful while revising their manuscript. With kind regards, Attilio Castellarin**

We would like to thank Attilio Castellarin for his positive feedback and further address his comments as follows.

**MAJOR CONCERNS**
**- Levee breaching is neglected**
**The authors clearly state that they neglected levee-breaching in their analysis. This is a rather strong assumption because it is far from real conditions (most likely breach) and because the formation of a breach has serious implication on the hazard downstream (lower flood peaks and volumes downstream the breach) and in the inundated area (larger outflow volumes if a breach is present relative to the no-breach case). I would suggest deepening the discussion on this main assumption.**

We thank the referee for this suggestion. We agree that neglected levee-breaching in the analysis is a strong assumption. In reality, dike breaches may lead to significant reductions of flood peaks downstream of breach locations and larger outflow volumes can be observed in the inundated area compare to the no-breach case. However, the modelling of levee-breaching requires high computational time. The prediction of breach locations is very difficult in practice; hence a stochastic approach including multiple Monte Carlo runs would be needed. In this study, the consideration of levee-breaching can increase the computational time where this is already high. We will expand the discussion in the revised manuscript accordingly. In addition, as we stated in page 15, line 371-372, "For this study, the model limitations are not seen as major concerns because the different assumptions on spatial dependence are investigated by using the same model.". Therefore, the assumption of neglected levee-breaching can be acceptable in this study.

**- Only direct losses to residential buildings**
**The authors apply a multiple-variable damage model that considers only direct damages to residential buildings. Given the strong expertise on flood damage modeling of the GFZ research team, I would like to see some discussion on this assumption as well. Based on their previous research activities, could the authors speculate if the same results of the study should apply also to other kind of damages (direct damages to the industrial/agricultural sector, indirect damages)? Are the three hypotheses on spatial dependence really interchangeable if indirect damages (e.g. disruption of services) are considered instead of direct ones?**

Our risk estimates in this study are based only on damages to residential buildings. If we consider other sectors like commercial or agricultural in the analysis, we will see an increase of exposed assets with higher monetary loss for all three hypotheses of spatial dependence. Therefore, we expect to see similar patterns of overestimation (underestimation) for large (small) return periods under complete dependence assumption for direct damage to other sectors, as well. Hence, we believe the exclusion of other sectors does not affect the final conclusions.
While direct flood damages occur due to the physical contact of the objects with the flood water, indirect damages occur outside the flood area, e.g. due to the interruption of public services or supply chains affecting the production of companies. The propagation of these effects depends strongly on the supply chains, i.e. economic links between

affected companies in and outside the flooded area. Therefore, the spatial inundation patterns and risk patterns would most likely be considerably different, strongly dependent on the economic system. Thus, including indirect damage into a spatial dependent study would require an in-depth economic analyses and would have a very different focus from the study we are presenting here.

**- Complete spatial dependence / independence**
**I believe I understood the technicalities for simulating flood damages and assessing flood risk under the hypotheses: (i) complete spatial dependence and (iii) complete independence. Yet, I would say that while the analysis framework is extremely clear for (ii) "modeled dependence" (perhaps also due to the flowchart of Fig. 2), In my opinion the description of the technicalities of how flood risk is modeled under hypotheses (i) and (iii) is not as clear (hydrological simulation is repeated, or streamflow time series are simply resampled?). This is a pity because the manuscript could serve as a blueprint for repeating the same modeling exercise in different areas or at larger scales (continental, global) considering all three spatial dependence typologies. Suggestion: it would be good to have an illustration on what differs in the flowchart of Fig. 2 when hypotheses (i) and (iii) are considered instead of (ii).**

We run regional flood model (RFM) and calculate economic damages over 10,000 years, just once. Without any resampling, these results represent (ii) modelled dependence assumption. Later, the other two assumptions (complete dependence and independence) are formed for different spatial units by using same damages at sub-basin level. Under the assumption of (i) complete dependence, T-year damage for certain spatial unit is represented by aggregating T-year flood damages at all sub-basins. Under the assumption of (iii) complete independence, damages at sub-basin level are randomly resampled for each flood event and then aggregated for the certain spatial unit. We will modify Figure 2 to express our approach clearer.

**SPECIFIC REMARKS**
**\* L.29 "there were 3062 floods", consider using a rounded figure, floods may occur without impacting people and therefore without being recorded**

We thank the reviewer. We will modify this statement as "there were around 3100 flood events".

**\* Fig. 2, consider enhancing readability (e.g. by using larger fonts)**

We will increase font size in this figure.

**\* Sections 3.1.1 and 3.1.2 consider commenting on the limitations associated with the simulation timescale (daily) relative to result in validation (poorer results in some small catchments)**

We appreciate very much for this constructive comment. The SWIM model for the Elbe was set up to model daily flood processes; therefore, the regional weather generator (RWG) provides daily time series of meteorological variables as input of the SWIM model. In general, SWIM model shows reasonably good results at daily time scale for the Elbe. However, relatively poor performance at a few small mountainous sub-basins, that react rather quickly, is observed. This fact will be added to the revised version of the manuscript.

**\* L.188, is the continuity of the levee crest ensured using a 10m res. DEM**

The continuity of the levee crest is ensured at many locations, however at few locations due to the uncertainty in DEM, we have used other information sources such as authorities.

**\*L. 193, consider including Manning's roughness units**

We will add the unit ($m^{-1/3}s$) of Manning's roughness to the manuscript.

**\* Fig.4, I am surprised by the remarkable narrowness of the confidence interval for the fully independent case, is it associated with the size of resampled sets? The authors should comment on this.**

We thank the reviewer for this technical comment. The resampling size of 1000 was used in the manuscript to estimate the complete independence curve (50th-percentile) and its confidence bounds (2.5th to 97.5th-percentile).

The plot below shows the estimation of 1000-year loss using three resampling size values of rs=10000, 1000 and 100. Density curves and their associated 95% mass bar with dots (the position of the bars with regard to x-axis has no meaning) illustrate that the resampling size of 1000 is sufficient to have reliable estimation. With this, we can confirm that for our case study, the confidence interval of the complete independence case is narrow as shown in the manuscript.

[Figure]

**\* Fig.6, panels could be reported (also) in a standardized form concerning the y-axes, so that the size of each region is neglected, the three curves are more complete in all panes and similarities/dissimilarities between different cases can be better illustrated**

We thank the referee for this suggestion. We will modify axes in Figure 6 to improve readability. However, the relation between the spatial scale and the risk curves under three assumptions is difficult to illustrate and it is not a linear relation. Because, the illustration of any standardized form for the y-axes may mislead the reader, we prefer to keep this figure as it is.